# Associations between pain and physical activity among older adults

**Nils Georg Niederstrasser** *, **Nina Attridge**

Department of Psychology, University of Portsmouth, Portsmouth, United Kingdom

* nils.niederstrasser@port.ac.uk

## Abstract

### Objectives

Chronic pain is a significant societal problem and pain complaints are one of the main causes of work absenteeism and emergency room visits. Physical activity has been associated with reduced risk of suffering from musculoskeletal pain complaints, but the exact relationship in an older adult sample is not known.

### Methods

Participants self-reported their physical activity level and whether they were often troubled by bone, joint, or muscle pain. Logistic regression analyses revealed the nature of the relationship between musculoskeletal pain and physical activity cross-sectionally and longitudinally over the course of 10 years. Data were taken from the English Longitudinal Study of Ageing, comprising of 5802 individuals residing in England aged 50 or older.

### Results

Only high levels of physical activity were associated with a reduced risk of suffering from musculoskeletal pain compared to a sedentary lifestyle longitudinally. In addition, having low wealth, being female, and being overweight or obese were found to be risk factors for suffering from musculoskeletal pain.

### Conclusions

The development of interventions aimed at alleviating and preventing musculoskeletal pain complaints might benefit from incorporating physical activity programs, weight loss, and aspects addressing wealth inequality to maximise their efficacy.

## Introduction

Chronic pain is related to considerable societal costs that stem from greater use of health care and reduced work productivity [1]. It presents one of the most widespread and complex problems in the medical community [2], as sufferers report decreased quality of life, and poor

**Data Availability Statement:** The ELSA dataset is freely available from the UK Data Service to all bonafide researchers. The dataset can be accessed here: https://discover.ukdataservice.ac.uk/series/?sn=200011.

**Funding:** The data were made available through the UK Data Archive. The English Longitudinal Study of Ageing was developed by a team of researchers based at University College London, NatCen Social Research, the Institute for Fiscal Studies and the University of Manchester. The data were collected by NatCen Social Research. The funding is provided by the National Institute of Aging (R01AG017644) and a consortium of UK government departments coordinated by the Economic and Social Research Council. Publication has been supported by the University of Portsmouth Open Access Fund. The funders had no role in the study design; in the collection, analysis, and interpretation of data; in writing of the report; or in the decision to submit the paper for publication. The developers and funders of ELSA and the Archive do not bear any responsibility for the analyses or interpretations presented here.

**Competing interests:** The authors have declared that no competing interests exist.

physical, social, and psychological well-being alongside their pain complaints [3, 4]. The prevalence of pain increases with age, with up to 62% of the over 75 age group in the UK reporting persistent pain complaints [5]. Perplexingly, however, there is an under-representation of older adults in pain clinics and pain management programmes [6]. Negative attitudes toward pain treatment, especially in terms of perceived lack of efficacy and concerns over adverse side effects as well as addiction, the belief in the inevitability of pain in old age, and pain complaints' perceived low importance compared to other comorbidities may explain why older adults are reluctant to seek treatment for pain [7]. In an ageing society, pain will pose an ever-increasing challenge. It is therefore imperative to identify what predisposes individuals to develop persistent pain complaints, so that interventions may be developed to prevent and reduce chronic pain.

Studies have suggested that sedentary behaviour may lead to disuse symptoms and result in greater pain sensitivity [8], while being physically active may have protective effects against the occurrence of pain and its consequences [9]. It is not clear, however, how these findings relate to older adults [10–12]. Older adults have a higher incidence of chronic pain [13] and are generally less physically active than younger adults [14]. Only 1 in 4 adults over 65 engage in the minimum recommended activity levels needed to maintain health [15], suggesting that sedentariness may be a major contributor to pain complaints among older adults.

Most examinations into the relationship between physical activity and pain have been cross-sectional [16, 17], with few longitudinal exceptions [18]. None, to date, have examined this relationship specifically in a large sample of older adults. The lack of longitudinal studies among older adults specifically is alarming, as potential risk factors, such as sedentariness, are more common but take time to lead to pain. This relationship may be obscured by cross-sectional investigations, as sedentary behaviour may be both a risk and a result of pain complaints. The impact of pain is often more severe among older adults [13] and it frequently occurs alongside multiple comorbidities further limiting treatment options and highlighting the need for preventative action [19]. Furthermore, up to 60% of care home residents suffer from some form of cognitive impairment that limits the ability to communicate pain, leading to both over and under-treatment of pain [20]. There is also a great level of variability in the assessment of both physical activity and pain, which may obscure more nuanced relationships [21] and so the exact nature of the long-term association between physical activity and pain among older adults remains elusive.

Furthermore, there may be additional factors associated with greater propensity to report pain complaints in older age. Several such factors have been suggested for the general population, including female gender [22, 23], being overweight [24], and wealth levels [25]. Persistent pain is more common among women than men [22, 23], which may be related to an increased sensitivity to pain that leads to more frequent widespread pain and higher reported pain intensity. Being overweight may cause excess mechanical stress predisposing individuals to develop persistent pain [24] and higher levels of proinflammatory cytokines that may produce a hyperalgesic state [26]. Economic stress and associated depressive symptomology stemming from low wealth levels may also predispose individuals towards developing pain complaints [25].

While there is evidence suggesting these factors may confound the relationship between physical activity and pain, their cumulative effects in a sample of older adults remain elusive. Here, data from the English Longitudinal Study of Ageing (ELSA) are used to examine whether physical activity, adjusting for gender, age, wealth level, and being overweight or obese, predicts the risk of reporting persistent musculoskeletal pain either cross-sectionally or longitudinally over the course of ten years.

## Methods

### Sample

ELSA is an ongoing longitudinal study that gathers data from a representative sample of adults aged 50 years and over living in England. Details pertaining to data collection methods and sampling details are available elsewhere [27]. Currently, data from nine waves are available, collected between 2002/2003 and 2020/2021, with two-year intervals between waves. The current study draws on data from waves 2 (2004/2005), 4, (2008/2009), 7 (2014/2015), and 9 (2018/2019). With each wave, new participants are added to maintain a steady sample. Waves 1 and 3 do not include information on relevant risk factors and so had to be excluded. Since the longitudinal assessment spans 10 years, participants present in either waves 2 and 7 or 4 and 9 were eligible for inclusion in the current study. In other words, some participants' data were collected between 2004/2005 and 2014/2015 and others' were collected between 2008/2009 and 2018/2019. Baseline data were taken from waves 2 (2004/2005) and 4 (2008/2009). Data were collected through a combination of nurse assessments and self-report measures. Participants gave fully informed written consent to participate in the study. The London Multicentre Research Ethics Committee (MREC/01/2/91) granted ethical approval for the data collection and archiving.

### Outcome—Ffrequent musculoskeletal pain

Participants self-reported whether they were often troubled by bone, joint, or muscle pain (yes/no) at waves 2, 4, 7, and 9. This variable has been used in previous investigations using the ELSA dataset [28].

### Physical activity level

Participants indicated the frequency of taking part in mild, moderate, and vigorous activities during leisure time, selecting from the following options for each level of intensity: (1) more than once per week, (2) once per week, (3) one to three times per month, or (4) hardly ever. Participants were shown prompt cards depicting examples of activities and corresponding intensity levels to aid them in finding the appropriate intensity level of their leisure activity. Examples of vigorous activity included: swimming, digging with a spade, jogging or running, cycling, and tennis; moderate: dancing, floor/stretching exercises, walking at a moderate pace, gardening and washing the car; mild: doing laundry, vacuuming, and home repairs.

The questions pertaining to physical activity status were taken from a validated physical activity interview [29]. Participants were categorised into four mutually exclusive groups (sedentary, mild, moderate, and vigorous), based on the highest intensity of physical activity that was carried out at least once per week [30]. For example, someone who did mild physical activity up to three times per month would be assigned to the "sedentary" group, while someone who did laundry every week and cycling once per month would be assigned to the "mild" group.

Participants were also asked to indicate the physical activity level of the work they do most of the time, with the options of sedentary occupation (most time spent sitting), standing occupation (most time spent standing or walking, but not requiring intense physical effort), physical work (requiring some physical effort such as handling of heavy objects and use of tools), and heavy manual work (requiring very vigorous physical activity including handling of very heavy objects).

Participants were assigned to categories of physical activity based on the following criteria:

Sedentary: Not working or sedentary occupation, engages in mild exercise 1–3 times a month or less, with no moderate or vigorous activity.

Low: Standing occupation, engages in moderate leisure-time exercise once a week or less and no vigorous activity; or engages in mild leisure-time activity at least 1–3 times a month, moderate once a week or less and no vigorous; or has a sedentary or no occupation and engages in moderate leisure-time activity once a week or 1–3 times a month, with no vigorous activity.

Moderate: Does physical work; or engages in moderate leisure-time activity more than once a week; or engages in vigorous activity once a week to 1–3 times a month.

High: Heavy manual work or vigorous leisure activity more than once a week.

## Covariates in the association between physical activity and persistent musculoskeletal pain

These additional predictor variables were assessed either during nurse visits or self-reported during data collection for waves 2 and 4.

**Body Mass Index (BMI).**   During nurse visits participants' standing height (meters) and body mass (kilograms) were measured. Participants exceeding a body mass of 130 kg were excluded from the measurement, as the scales had a maximum weight capacity of 130 kg. In these cases, body mass was estimated. Body mass index was quantified as dividing body mass (kg) by standing height (meters) squared ($kg/m^2$). The following categories were used: BMI < 18.5 underweight; BMI between 18.5 and 25 normal weight; BMI between 25 and 30 overweight; BMI > 30 obese.

**Age.**   Ages over 90 were collapsed into a single age group, to protect participants' identities.

**Sex.**   Sex was self-reported by participants during interviews.

**Wealth quintile.**   Wealth was determined by dividing participants into quintiles based on their net wealth. Net wealth was quantified as the net sums of housing wealth, physical wealth (including additional property wealth, wealth related to business and other physical assets) and financial wealth (including savings, stock certificates and bank accounts) after financial debt and mortgage debt had been subtracted.

## Statistical approach

All analyses were performed using R version 4.1. The outcome variable "Musculoskeletal Pain" was quantified as binary (*are you often troubled by pain*: "yes"/ "no") and three logistic regression analyses, using the R-package "glm", were used to assess the relationship between physical activity and musculoskeletal pain a) cross-sectionally and b) longitudinally.

**Cross-sectional analysis.**   We ran a logistic regression analysis with musculoskeletal pain at baseline as the dependent variable and age, physical activity level, BMI, gender, and wealth quintile as predictor variables.

**Longitudinal analyses.**   We ran two logistic regression analyses with musculoskeletal pain at follow up (10 years after baseline) as the dependent variable and age, physical activity level, BMI, gender, wealth quintile, and musculoskeletal pain at baseline as predictor variables. The first analysis comprised of the entire data. The second analysis included only those reporting to be pain free at baseline and therefore musculoskeletal pain at baseline was not used as a predictor variable in this analysis.

Odds ratios for the continuous and each level of the categorical predictors in each analysis were calculated.

In cases where data for participants were available for the ten years following wave 2 and 4, data from wave 2 were used as baseline. Wherever data were available only from wave 4, these were used as baseline. Outcome data were taken from waves 7 and 9 based on which wave was used as baseline.

## Results

### Sample characteristics

In total, data from 5802 individuals (mean age 62.3, *SD* = 7.7; 2559 (44.1%) males), were available over a period of ten years. Table 1 presents the means and standard deviations as well as counts for the categorical variables.

### Cross-sectional relationship between physical activity and pain

At baseline, 2062 participants reported being troubled by musculoskeletal pain. Participants' age, physical activity level, BMI, gender, and wealth quintile were entered into the regression equation in one step. Values from the final regression equation (see Table 2) indicated that mild, moderate, and high physical activity were associated with a lower likelihood of suffering from musculoskeletal pain compared to being sedentary. Similarly, belonging to a higher

**Table 1. Sample overview.**

| Variables | | |
|---|---|---|
| Troubled by musculoskeletal pain | **Yes** | No |
| Baseline | 2062 (35.5%) | 3740 (64.5%) |
| 10-year follow-up | **Yes** | No |
| | 2461 (42.4%) | 3341 (57.6%) |
| Sex | **Male** | Female |
| | 2559 (44.1%) | 3243 (55.9%) |
| Age—mean (SD) | 62.3 (7.7) | |
| Physical activity level (n) | | |
| **Sedentary** | 122 (2.1%) | |
| Mild | 1124 (19.4%) | |
| Moderate | 3137 (54.1%) | |
| High | 1419 (25.5%) | |
| BMI categories (n) | | |
| Underweight | 34 (0.6%) | |
| **Normal** | 1547 (26.7%) | |
| Overweight | 2496 (43.0%) | |
| Obese | 1725 (29.7%) | |
| Wealth Quintiles | | |
| **Low** | 751 (12.9%) | |
| Low to Medium | 1005 (17.3%) | |
| Medium | 1158 (20.0%) | |
| Medium to High | 1330 (22.9%) | |
| High | 1558 (26.9%) | |

N = 5802; unless otherwise indicated, variables refer to those taken at baseline;

for categorical variables reference categories are printed in bold;

**Table 2. Logistic regression for the cross-sectional relationship between physical activity and musculoskeletal pain.**

|  | β (SE) | Odds Ratio (95% Confidence Interval) |
|---|---|---|
| Intercept | 1.27 (0.30) | |
| Age | 0.01 (0.00) | 1.01 (1.00–1.01) |
| Reference category: sedentary PA | | |
| mild PA | -0.72 (0.21)** | 0.49 (0.32–0.74) |
| moderate PA | -1.39 (0.21)** | 0.25 (0.16–0.37) |
| high PA | -1.58 (0.22)** | 0.21 (0.13–0.31) |
| Reference category: normal weight | | |
| underweight | 0.04 (0.39) | 1.04 (0.47–2.16) |
| overweight | 0.23 (0.07)** | 1.26 (1.09–1.46) |
| obese | 0.65 (0.08)** | 1.91 (1.64–2.23) |
| Reference category: male gender | | |
| female gender | 0.28 (0.06)** | 1.33 (1.18–1.49) |
| Reference category: Low Income | | |
| Low to Middle | -0.40 (0.10)** | 0.67 (0.55–0.82) |
| Middle | -0.58 (0.09)** | 0.56 (0.46–0.68) |
| Middle to High | -0.67 (0.10)** | 0.51 (0.42–0.62) |
| High | -0.74 (0.10)** | 0.48 (0.39–0.57) |

N = 5802;

$^*$ $p < .05$;

$^{**}$ $p < .01$;

Beta weights are from the final regression equation;

$R^2$ for final regression equation = .11 (Nagelkerke), Model $x^2$(12) = 461.68, $p < .01$;

wealth quintile was associated with being less likely to suffer from musculoskeletal pain. Being female, overweight or obese were risk factors associated with an increased likelihood of suffering from musculoskeletal pain complaints.

## Longitudinal relationship between physical activity and pain

Next, we examined the association between physical activity and musculoskeletal pain in the 10 years following baseline. The first analysis included all 5802 participants. Participants' age, physical activity level, BMI, gender, wealth quintile, and pain status at baseline were entered into a logistic regression equation in one step, with pain status at follow up as the dependent variable. Out of 5802 participants, 2461 participants reported being frequently troubled by musculoskeletal pain ten years after baseline.

Beta weights for the final regression equation (see Table 3) indicated that engaging in high physical activity was associated with lower probability of reporting being troubled by musculoskeletal pain ten years later. Participants in the highest three wealth quintiles were also less likely to report suffering from musculoskeletal pain compared to those in the lowest quintile. Being female, overweight, or obese was associated with an increased the risk of suffering from musculoskeletal pain complaints after ten years.

Finally, we examined the influence of physical activity on developing musculoskeletal pain in the 10 years following baseline by examining only participants who reported being free at baseline (n = 3704). Pain status at follow up was used as the dependent variable and age, physical activity level, BMI, gender, and wealth quintile were entered in one step as predictors into

**Table 3. Logistic regression for the longitudinal relationship between physical activity and musculoskeletal pain.**

| | β (SE) | Odds Ratio (95% Confidence Interval) |
|---|---|---|
| Intercept | 1.27 (0.30) | |
| Reference category: No current musculoskeletal pain | | |
| Existing musculoskeletal pain | 1.53 (0.06)** | 4.60 (4.01–5.20) |
| Age | -0.00 (0.00) | 1.00 (1.00–1.01) |
| Reference category: sedentary PA | | |
| mild PA | -0.16 (0.22) | 0.85 (0.55–1.30) |
| moderate PA | -0.32 (0.22) | 0.72 (0.47–1.10) |
| high PA | -0.52 (0.22)* | 0.59 (0.38–0.91) |
| Reference category: normal weight | | |
| underweight | -0.05 (0.40) | 0.95 (0.43–2.03) |
| overweight | 0.29 (0.07)** | 1.34 (1.16–1.55) |
| obese | 0.61 (0.08)** | 1.85 (1.58–2.16) |
| Reference category: male gender | | |
| female gender | 0.44 (0.06)** | 1.55 (1.38–1.75) |
| Reference category: Low Income | | |
| Low to Middle | -0.18 (0.11) | 0.84 (0.68–1.04) |
| Middle | -0.46 (0.11)** | 0.63 (0.51–0.78) |
| Middle to High | -0.38 (0.10)** | 0.68 (0.56–0.83) |
| High | -0.51 (0.10)** | 0.60 (0.49–0.73) |

$R^2$ for final regression equation = .23 (Nagelkerke), Model $x^2$(13) = 1079.50, $p < .01$;

N = 5802;

* $p < .05$;

** $p < .01$;

Beta weights are from the final regression equation;

the logistic regression equation. Out of 3704 participants, 1058 participants reported being frequently troubled by musculoskeletal pain ten years after baseline.

It can be seen from the beta weights for the final regression equation (see Table 4) that engaging in high levels of physical activity was associated with a reduced risk of developing pain complaints after 10 years. Participants in the highest three wealth quintiles were also less likely to develop musculoskeletal pain compared to those in the lowest quintile. Being female, overweight, or obese increased the risk of suffering from musculoskeletal pain complaints after ten years.

## Discussion

This study found that physical activity has a beneficial influence on musculoskeletal pain complaints in a sample of older adults living in England, both cross-sectionally and longitudinally. This relationship existed over and above the influence of age, weight, gender, and wealth on pain. Being overweight or obese, female, having existing musculoskeletal pain, and having low wealth were found to predispose individuals to report frequent musculoskeletal pain complaints 10 years later. These factors were all significant independently of each other. The current study is the first to examine the concurrent contributions of these variables to the risk of experiencing musculoskeletal pain complaints cross-sectionally and longitudinally in a large sample of older adults.

In our cross-sectional analysis, all levels of physical activity were found to be associated with a lower risk of reporting being troubled by musculoskeletal pain, over and above gender,

**Table 4. Logistic regression for the longitudinal relationship between physical activity and musculoskeletal pain among those reporting no pain at baseline.**

| | β (SE) | Odds Ratio (95% Confidence Interval) |
|---|---|---|
| Intercept | -1.10 (0.50) | |
| Age | 0.01 (0.01) | 1.00 (1.00–1.02) |
| Reference category: sedentary PA | | |
| mild PA | -0.56 (0.36) | 0.57 (0.28–1.17) |
| moderate PA | -0.64 (0.35) | 0.53 (0.26–1.06) |
| high PA | -0.81 (0.36)* | 0.45 (0.22–0.91) |
| Reference category: normal weight | | |
| underweight | -0.40 (0.56) | 0.67 (0.19–1.82) |
| overweight | 0.33 (0.09)** | 1.39 (1.16–1.66) |
| obese | 0.62 (0.10)** | 1.85 (1.52–2.26) |
| Reference category: male gender | | |
| female gender | 0.43 (0.08)** | 1.54 (1.33–1.79) |
| Reference category: Low Income | | |
| Low to Middle | -0.03 (0.14) | 0.98 (0.74–1.29) |
| Middle | -0.30 (0.14)* | 0.74 (0.56–0.98) |
| Middle to High | -0.31 (0.14)* | 0.74 (0.56–0.96) |
| High | -0.40 (0.14)** | 0.67 (0.52–0.88) |

N = 3740;

* $p < .05$;

** $p < .01$;

Beta weights are from the final regression equation;

$R^2$ for final regression equation = .05 (Nagelkerke), Model $x^2(12)$ = 118.23, $p < .01$;

weight, and wealth. Longitudinally, however, only high physical activity was associated with a lower probability of reporting being troubled by musculoskeletal pain ten years later. Physical activity has a beneficial effect on weight and may further improve bone mass and muscle function, prevent falls, as well as improve general health [31]. Further, physical activity may positively impact pain by elevating mood [32], reducing stress [33], and enhancing descending pain modulation [17]. Some investigations suggest a U-shaped relationship between physical activity and pain [21], whereby low and high levels of activity are related to increased pain, suggesting it can be both a risk and preventative factor. The current study did not find a U-shaped relationship between pain and physical activity; in the cross-sectional analysis all levels of physical activity were associated with a lower risk of pain than being sedentary, and in the longitudinal analysis high physical activity was associated with a lower risk of reporting musculoskeletal pain than being sedentary. However, the measure of physical activity used in this study precludes identification of those who engage in excessive levels of exercise and may therefore be at higher risk of injury and pain. Furthermore, participants' relatively old age may be associated with reduced exercise intensity and frequency, further obscuring the proposed U-shaped relationship. Nevertheless, the current study suggests that engaging in high physical activity may be beneficial in preventing or alleviating musculoskeletal pain complaints.

Current musculoskeletal pain complaints were the strongest predictor of future pain, suggesting that participants may suffer from pain for extended periods of time. Chronic pain is defined as pain lasting for more than three months [34], but the majority suffer for longer periods. A study among European chronic pain patients found that 59% of respondents had suffered from pain for between 2 and 15 years [22]. The reported association between existing

and future musculoskeletal pain complaints is therefore not surprising and only serves to highlight the importance of adequately identifying, treating, and preventing pain.

The current investigation corroborates findings that women are at higher risk of reporting being troubled from musculoskeletal pain [22, 23]. The exact mechanisms underlying gender differences in pain are unknown but have been hypothesised to be an interaction between biological, psychological, and sociocultural factors [35]. Differences in levels of sex hormones, such as oestrogen, progesterone, and testosterone, may contribute to the marked sex-related differences in pain [35–37]. Hormonal differences may also affect the processing of pain-related stimuli in the brain, with women showing lower activation in pain inhibitory brain regions [38]. Differences in psychosocial aspects, such as pain catastrophizing and self-efficacy, have also been suggested to mediate sex differences in pain responsivity. Women tend to engage in more catastrophic thinking around pain and have lower self-efficacy compared to men [35, 39], which is associated with greater pain. Finally, sociocultural beliefs pertaining to femininity and masculinity may influence pain reporting [35]. Reporting pain is societally more accepted among women than men, which may explain part of the observed differences.

There is currently no consensus as to the exact nature of the relationship between pain and obesity [40]. The present study's longitudinal nature lends credence to the notion that obesity is a precursor to musculoskeletal pain, but the reverse cannot be excluded based on these data, as obesity itself was not manipulated. There are two main hypotheses explaining the effect of obesity on pain [40]. First, it has been hypothesised that joint degradation due to excessive body mass may predispose individuals to develop osteoarthritis, especially in the lower back, knee, and hip joints. Second, obese individuals tend to have higher levels of certain inflammatory markers, such as tumour necrosis factor α, interleukin 6, and C-reactive protein, which may produce a hyperalgesic state [26]. Alternatively, pain may also predispose individuals to become obese because of reduced activity. A sedentary lifestyle, due to pain complaints, may lead to a calorie surplus, which may be negatively reinforced by the analgesic effect of food consumption [40]. Nevertheless, what the present study shows is that obesity is an important factor when it comes to musculoskeletal pain. Regardless of the directionality of the causal link, reducing obesity is likely to have a positive effect not only on musculoskeletal pain but on a range of other health-related aspects, such as the cardiovascular system. Furthermore, it is reasonable to assume that reductions in pain may serve to increase individuals' inclination to exercise, further reducing obesity likely having overall positive effects on health.

Wealth was associated with a lower risk of musculoskeletal pain complaints. The negative effects on mood and general health stemming from economic stress are well documented [41]. Depression and pain have overlapping neuroanatomical pathways and share neurobiological substrates, which may explain why depressive symptomologies are associated with pain [42, 43] and ultimately why low wealth is associated with musculoskeletal pain complaints [25]. Furthermore, a high disposable income may also enable individuals to seek extra care, aside from that covered by insurances or national health services, to treat ailments. On the other hand, pain is a major contributor to work absenteeism, reduced employability, and loss of employment. All of which are associated with a loss of income and therefore negatively affect individuals' wealth. Given the nature of the data, the directionality of this effect cannot be elucidated.

The current study's strengths include a large representative sample of older adults residing in England, a follow-up period of 10 years, as well as a range of self-reported and objective predictor variables. The outcome variable was a self-report of whether participants were frequently troubled by pain. As such, responses may have been influenced by recall bias and social desirability. Furthermore, use of a dichotomous variable does not allow inferences to be drawn as to the predictors of pain intensity. Nevertheless, the same outcome variable has been

used in previous investigations, such as Smith et al. [28]. This research was concerned specifically with the association between physical activity and pain in older adults (aged 50+) and it should therefore be noted that the conclusions may not apply to younger adults. Furthermore, self-reported physical activity levels may be subject to recall bias, which may lead to misclassification.

In conclusion, the aim of this study was to examine the relationship between physical activity and musculoskeletal pain complaints in a sample of older adults. It showed that high physical activity is associated with a reduced likelihood of developing musculoskeletal pain complaints compared to sedentariness, over and above age, weight, gender, and wealth. These findings provide insights that may inform interventions aimed at reducing the risk of developing frequent musculoskeletal pain complaints. In particular, weight control, increasing physical activity and managing wealth inequality should be considered when developing preventative strategies to reduce pain.

## Author Contributions

**Conceptualization:** Nils Georg Niederstrasser, Nina Attridge.

**Data curation:** Nils Georg Niederstrasser.

**Formal analysis:** Nils Georg Niederstrasser, Nina Attridge.

**Methodology:** Nils Georg Niederstrasser.

**Writing – original draft:** Nils Georg Niederstrasser, Nina Attridge.

**Writing – review & editing:** Nils Georg Niederstrasser, Nina Attridge.

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
