## [Decision Letter · Decision Letter 0]

30 Dec 2021

PONE-D-21-34233Physical activity protects against pain in older adultsPLOS ONE

Dear Dr. Niederstrasser,

Thank you for submitting your manuscript to PLOS ONE. After careful consideration, we feel that it has merit but does not fully meet PLOS ONE’s publication criteria as it currently stands. Therefore, we invite you to submit a revised version of the manuscript that addresses the points raised during the review process.Please ensure that your decision is justified on PLOS ONE’s publication criteria and not, for example, on novelty or perceived impact.

We look forward to receiving your revised manuscript.

Kind regards,

David Meyre

Academic Editor

PLOS ONE

Journal Requirements:

Reviewers' comments:

Reviewer's Responses to Questions

**Comments to the Author**

1. Is the manuscript technically sound, and do the data support the conclusions?

Reviewer #1: Yes

Reviewer #2: Yes

2. Has the statistical analysis been performed appropriately and rigorously? 

Reviewer #1: Yes

Reviewer #2: Yes

3. Have the authors made all data underlying the findings in their manuscript fully available?

Reviewer #1: Yes

Reviewer #2: Yes

4. Is the manuscript presented in an intelligible fashion and written in standard English?

Reviewer #1: Yes

Reviewer #2: Yes

5. Review Comments to the Author

Reviewer #1: I would like to extend my congratulations to the authors. DEal with huge datasets is not a simple task and you should be proud of your work.

The paper is well written and present relevant data; however, the conclusions seem to be a little overestimated.

For instance, on page 9 the authors state that, on a cross-sectional point of view, being physically active decreases the risk of experiencing musculoskeletal pain and on page 10, on a longitudinal point analysis, being physically active decreases the risk of experiencing musculoskeletal pain, however, the OR presented on table 3 regarding the longitudinal analysis only support this conclusion for those engaging in vigorous physical activity.

The same goes for the discussion section, where authors lump all the results together as if all the analysis were statistically significant. These should be addressed more carefully to fully describe the finding, eliminating the chance of misleading the reader.

Also, I suggest that in the discussion section the authors should comment about their subjects’ demographics, specifically regarding the advanced age and above-average wealth status of the subjects. These could be major confounding factors and should be acknowledge on the paper.

Reviewer #2: The study addressed a very important question about physical activity and musculoskeletal pain using a large dataset. However, the aims of the paper may not be consistent with the title. It seems like the paper is looking at multiple factors instead of PA only.

1. Abstract: the information about how pain and PA were assessed were lacking. What is the definition of "no physical activity"?

2. Intro: 1st paragraph, which country were you referring to when you mentioned the prevalence of pain? Is this the prevalence for community-dwelling older people or other settings?

3. Intro: the last sentence in the 1st paragraph may not smoothly transit to the second paragraph. It sounds like you are looking at factors contributing to persistent pain complaints and then the next paragraph is about the protective effects of PA on pain.

4. The second paragraph on page 4, you used the term "interact", it sounds like you are examining the interaction between PA and other factors which is not in your analyses. Again, is the evidence you provided specifically for older adults? This needs to be clear.

5. After I read the intro, I think this paper is focusing on PA and pain because you have mentioned a lot about this association, but in the results and discussions, it seems like you are looking at multiple factors contributing to pain.

6. Methods: the sample section is not clear. You said between waves 1 and 4 were eligible, but before and after that sentence, you only mentioned waves 2 and 4.

7. In terms of the pain and PA assessment, what is the time frame? Is this self-reported pain in the past year or past month? The same question for PA assessment. Older adults may not be able to accurately recall pain or PA if it is a long time frame.

8. Results: page 9, the word "increase" implies longitudinal association. Better to say "higher level of PA was associated with lower likelihood of reporting being troubled...."

9. Table 2, given the small beta coefficient for age, better to scale it to 5 or 10 years.

10. Results: page 10, "our analysis corrected for musculoskeletal pain complaints existing at baseline". The word "corrected" may not be accurate, better to say "adjusted for baseline pain". The sentence "the more PA an individual engaged in at baseline the less likely they were..." is not reflecting the results in table 3. The only significant finding is vigorous PA versus sedentary PA. The interpretation of the results should be accurate.

11. Discussion: second paragraph, the first sentence is too ambitious. This study is observational and not an intervention. The associations did not translate to "protect effects". In addition, better to add discussion about potential pathways from PA to pain in this paragraph. The last sentence "the current study suggests that all levels of physical activity are beneficial..." is not reflecting the results. Based on Table 3, it is not all levels of PA, only vigorous PA is associated with lower likelihood of pain.

12. Limitation: may add limitation of self-reported PA which leads to recall bias and misclassification.

13. Overall, the author mentioned the outcome is "development of persistent pain" which implies changes in pain status. I don't think the analysis simply adjusting for baseline pain is answering this question. Better to categorize participants into groups: no pain in both visits, incident pain (no pain at baseline), recovered pain, and persistent pain (pain in both visits).

6. PLOS authors have the option to publish the peer review history of their article (what does this mean?). If published, this will include your full peer review and any attached files.

Reviewer #1: **Yes: **DANIEL POZZOBON

Reviewer #2: No

---

## [Author Response · Author response to Decision Letter 0]

11 Jan 2022

Reviewer #1: I would like to extend my congratulations to the authors. Deal with huge datasets is not a simple task and you should be proud of your work.

Thank you very much for the kind words. We appreciate the support.

The paper is well written and present relevant data; however, the conclusions seem to be a little overestimated.

For instance, on page 9 the authors state that, on a cross-sectional point of view, being physically active decreases the risk of experiencing musculoskeletal pain and on page 10, on a longitudinal point analysis, being physically active decreases the risk of experiencing musculoskeletal pain, however, the OR presented on table 3 regarding the longitudinal analysis only support this conclusion for those engaging in vigorous physical activity.

The same goes for the discussion section, where authors lump all the results together as if all the analysis were statistically significant. These should be addressed more carefully to fully describe the finding, eliminating the chance of misleading the reader.

Thank you for pointing out these oversights. We have adjusted the text following your suggestions (“Beta weights for the final regression equation (see Table 3) indicated that engaging in mild, moderate, and high physical activity was protective against developing musculoskeletal pain complaints.”, pg. 12).

We have amended the discussion section in various places following the reviewer’s suggestions (“Longitudinally, however, only high physical activity was associated with a lower probability of reporting being troubled by musculoskeletal pain ten years later.” pg. 14; “Nevertheless, the current study suggests that engaging in high physical activity may be beneficial in preventing or alleviating musculoskeletal pain complaints.”, pg. 15).

Also, I suggest that in the discussion section the authors should comment about their subjects’ demographics, specifically regarding the advanced age and above-average wealth status of the subjects. These could be major confounding factors and should be acknowledge on the paper.

We have added a section explaining the possible confounding influence of participants’ advanced age and the self-report assessment (“This research was concerned specifically with the association between physical activity and pain in older adults (aged 50+) and it should therefore be noted that the conclusions may not apply to younger adults. Furthermore, self-reported physical activity levels may be subject to recall bias, which may lead to misclassification.”, pg. 17). While wealth is known to be associated with both physical activity and pain, as our paper also demonstrates, it is not necessarily the case that the participants selected in this study have a higher wealth status than average. The ELSA data set is specifically designed to be representative of older adults living in England and wealth distribution within the data set should therefore also be representative of wealth distribution in England among this demographic.

 

Reviewer #2: The study addressed a very important question about physical activity and musculoskeletal pain using a large dataset. However, the aims of the paper may not be consistent with the title. It seems like the paper is looking at multiple factors instead of PA only.

We thank reviewer 2 for their comments and suggested revisions. We have taken these on board and revised the manuscript to improve clarity. We feel the paper has been greatly improved following revisions based on suggestions by reviewer 2.

1. Abstract: the information about how pain and PA were assessed were lacking. What is the definition of "no physical activity"?

Thank you for pointing out these oversights in the abstract. We have added information regarding how pain and physical activity were assessed (“Participants self-reported their physical activity level and whether they were often troubled by bone, joint, or muscle pain”, pg. 2).

There was in fact no category denoting total absence of physical activity. The category reflecting the lowest level of physical activity was “sedentary”. The abstract has been changed to reflect this correction. The definition of “sedentariness” in the current context was as follows: Not working or sedentary occupation, engages in mild exercise 1–3 times a month or less, with no moderate or vigorous activity. We have added definitions of physical activity levels to the method section (pgs. 6-7). We have also changed the highest level of physical activity level from “vigorous” to “high” to more closely reflect the nomenclature used in the ELSA documentation.

2. Intro: 1st paragraph, which country were you referring to when you mentioned the prevalence of pain? Is this the prevalence for community-dwelling older people or other settings?

The figure refers to patient populations in US nursing homes and females in developing countries. We realise it may be more appropriate to report values from a UK-based population which are more comparable to the current study’s population. The section has been amended accordingly (“The prevalence of pain increases with age, with up to 62% of the over 75 age group in the UK reporting persistent pain complaints 2”, pg. 3).

3. Intro: the last sentence in the 1st paragraph may not smoothly transit to the second paragraph. It sounds like you are looking at factors contributing to persistent pain complaints and then the next paragraph is about the protective effects of PA on pain.

We have amended the starting sentence of the second paragraph to begin with the negative effects of sedentariness (“Studies have suggested that sedentary behaviour may lead to disuse symptoms and result in greater pain sensitivity 3, while being physically active may have protective effects against the occurrence of pain complaints and its consequences 1.”, pg. 3).

4. The second paragraph on page 4, you used the term "interact", it sounds like you are examining the interaction between PA and other factors which is not in your analyses. Again, is the evidence you provided specifically for older adults? This needs to be clear.

We have removed the word “interact” and changed the sentence’s wording (“Furthermore, there may be additional factors associated with greater propensity to report pain complaints in older age”, pg. 4). We have further clarified that the presented evidence is for the general population and not older adults specifically.

5. After I read the intro, I think this paper is focusing on PA and pain because you have mentioned a lot about this association, but in the results and discussions, it seems like you are looking at multiple factors contributing to pain.

The reviewer is correct in stating that the paper’s main focus is on the association between physical activity and pain. We feel, however, that it is important to appreciate alternative influences and factors when examining this association. We therefore decided to include additional factors in our analysis to show that physical activity predicts pain over and above the influences of these other variables. We have clarified at the beginning and end of the discussion: 

“This study found that physical activity has a beneficial influence on musculoskeletal pain complaints in a sample of older adults living in England, both cross-sectionally and longitudinally. This relationship existed over and above the influence of age, weight, gender and wealth on pain.” (p. 14) and “In conclusion, the aim of this study was to examine the relationship between physical activity and musculoskeletal pain complaints in a sample of older adults. It showed that high physical activity is associated with a reduced likelihood of developing musculoskeletal pain complaints compared to sedentariness, over and above age, weight, gender and wealth.” (p. 18).

6. Methods: the sample section is not clear. You said between waves 1 and 4 were eligible, but before and after that sentence, you only mentioned waves 2 and 4.

We realise the sample section is not written with sufficient clarity and have amended the section to rectify this. In short, waves 1 and 3 could not function as baseline assessments as they lacked essential variables (recorded variables vary between waves in the ELSA data set). Participants were eligible for inclusion if their data were present in either waves 2 and 7 or 4 and 9, since the longitudinal assessment spanned 10 years.

7. In terms of the pain and PA assessment, what is the time frame? Is this self-reported pain in the past year or past month? The same question for PA assessment. Older adults may not be able to accurately recall pain or PA if it is a long time frame.

There is no timeframe specified in the question assessing physical activity, according to the official ELSA documentation. The questions on physical activity status were extracted from a validated physical activity interview and have been used previously in the HSE physical activity assessment and other work using the data longitudinally 7. Similarly, there is no specific timeframe attached to the question regarding pain.

8. Results: page 9, the word "increase" implies longitudinal association. Better to say "higher level of PA was associated with lower likelihood of reporting being troubled...."

Thank you for this suggestion. We have adjusted the paragraph to reflect this change.

9. Table 2, given the small beta coefficient for age, better to scale it to 5 or 10 years.

Our understanding of this point is that the reviewer would like us to change the continuous variable of age into an ordinal variable, whereby participants are put into age groups such as 50-59, 60-69, etc. Our preference is to keep the variable continuous because this maximises its variance and thus information value. We do not see the small beta coefficient as surprising or problematic because the sample is specifically of older adults. If the sample had a wider age range, then we would expect to see a relationship between age and presence/absence of pain. 

10. Results: page 10, "our analysis corrected for musculoskeletal pain complaints existing at baseline". The word "corrected" may not be accurate, better to say "adjusted for baseline pain". The sentence "the more PA an individual engaged in at baseline the less likely they were..." is not reflecting the results in table 3. The only significant finding is vigorous PA versus sedentary PA. The interpretation of the results should be accurate.

We have replaced “corrected” with “adjusted”, as requested.

Thank you for pointing out the oversight regarding the effects of vigorous physical activity. This has been corrected (“Beta weights for the final regression equation (see Table 3) indicated that engaging in high physical activity was associated with lower probability of reporting being troubled by musculoskeletal pain ten years later.”, pg. 12).

11. Discussion: second paragraph, the first sentence is too ambitious. This study is observational and not an intervention. The associations did not translate to "protect effects". 

In addition, better to add discussion about potential pathways from PA to pain in this paragraph. The last sentence “the current study suggests that all levels of physical activity are beneficial…” is not reflecting the results. Based on Table 3, it is not all levels of PA, only vigorous PA is associated with lower likelihood of pain.

We have amended the first sentence in the second paragraph of the discussion to reflect the findings more accurately (“In our cross-sectional analysis, all levels of physical activity were found to be associated with a lower risk of reporting being troubled by musculoskeletal pain, over and above gender, weight, and wealth. Longitudinally, however, only high physical activity was associated with a lower probability of reporting being troubled by musculoskeletal pain ten years later.” pg. 14).

We have added a brief discussion of potential pathways from physical activity to pain to the paragraph (“Further, physical activity may positively impact pain by elevating mood 6, reducing stress 4, and enhancing descending pain modulation 5.”, pgs. 14-15). 

We have corrected the statement regarding all levels of physical activity being associated with lower likelihood of reporting pain complaints longitudinally.

We have also adjusted the manuscript’s title to reflect the changes.

12. Limitation: may add limitation of self-reported PA which leads to recall bias and misclassification.

The requested additions have been made (“Furthermore, self-reported physical activity levels may be subject to recall bias, which may lead to misclassification.” pg. 17).

13. Overall, the author mentioned the outcome is "development of persistent pain" which implies changes in pain status. I don't think the analysis simply adjusting for baseline pain is answering this question. Better to categorize participants into groups: no pain in both visits, incident pain (no pain at baseline), recovered pain, and persistent pain (pain in both visits).

We understand the reviewer’s concerns about whether our analysis explores the development of persistent pain or not. We have therefore added an additional analysis to the paper where we predict pain at Time 2 in the subset of participants who did not report pain at Time 1. We feel that this looks at the issue of the development of pain more specifically. However, we have retained our original analysis where we adjusted for baseline pain as we feel that together these analyses build a broader (and consistent) picture of the relationships. We feel that this addresses the research question more directly than using the four groups the reviewer suggests as a dependent variable.

We appreciate the reviewer suggesting an alternative analysis. We are, however, not aware of an analysis more appropriate than the current to demonstrates the effect we describe in the manuscript. We would appreciate if the reviewer could elaborate on the requested analysis.

 

References:

1. Bergman S: Public health perspective - how to improve the musculoskeletal health of the population. Best Pract Res Clin Rheumatol [Internet] 21:191–204, 2007 [cited 2021 Jun 30]. Available from: https://www.sciencedirect.com/science/article/pii/S1521694206001161?casa_token=50MwAUpTJnkAAAAA:f2qXyUAkuRUYbYzRWW6X6uNa95P30cmva_5uI95qsJEbj0w77ryl7i9pcc8RP2qhNwM0iwzF

2. Fayaz A, Croft P, Langford RM, Donaldson LJ, Jones GT: Prevalence of chronic pain in the UK: A systematic review and meta-analysis of population studies. BMJ Open [Internet] 6:, 2016 [cited 2022 Jan 7]. Available from: https://bmjopen.bmj.com/content/6/6/e010364?cpetoc

3. Law LF, Sluka KA: How does physical activity modulate pain? Pain. 2017. 

4. Mücke M, Ludyga S, Colledge F, Gerber M: Influence of Regular Physical Activity and Fitness on Stress Reactivity as Measured with the Trier Social Stress Test Protocol: A Systematic Review. Sport Med Springer International Publishing; 48:2607–22, 2018. 

5. Naugle KM, Riley JL: Self-reported physical activity predicts pain inhibitory and facilitatory function. Med Sci Sports Exerc [Internet] NIH Public Access; 46:622–9, 2014 [cited 2021 Jul 20]. Available from: /pmc/articles/PMC3906218/

6. Penedo FJ, Dahn JR: Exercise and well-being: A review of mental and physical health benefits associated with physical activity. Curr Opin Psychiatry [Internet] Curr Opin Psychiatry; 18:189–93, 2005 [cited 2022 Jan 6]. Available from: https://pubmed.ncbi.nlm.nih.gov/16639173/

7. Rogers NT, Marshall A, Roberts CH, Demakakos P, Steptoe A, Scholes S: Physical activity and trajectories of frailty among older adults: Evidence from the English Longitudinal Study of Ageing. Ginsberg SD, editor. PLoS One [Internet] 12:e0170878, 2017 [cited 2018 May 30]. Available from: http://dx.plos.org/10.1371/journal.pone.0170878

---

## [Editor Report · Decision Letter 1]

18 Jan 2022

Associations between Pain and Physical Activity among older Adults

PONE-D-21-34233R1

Dear Dr. Niederstrasser,

We’re pleased to inform you that your manuscript has been judged scientifically suitable for publication and will be formally accepted for publication once it meets all outstanding technical requirements.

Kind regards,

David Meyre

Academic Editor

PLOS ONE
---

## [Editor Report · Acceptance letter]

21 Jan 2022

PONE-D-21-34233R1 

Associations between Pain and Physical Activity among older Adults 

Dear Dr. Niederstrasser:

I'm pleased to inform you that your manuscript has been deemed suitable for publication in PLOS ONE. Congratulations! Your manuscript is now with our production department. 

Kind regards, 

on behalf of

Dr. David Meyre 

Academic Editor

PLOS ONE